# A Feasibility Study of the Use of Smartwatches in Wearable Fall Detection Systems

**DOI:** 10.3390/s21062254

**Published:** 2021-03-23

**Authors:** Francisco Javier González-Cañete, Eduardo Casilari

**Affiliations:** Departamento de Tecnología Electrónica, Universidad de Málaga, ETSI Telecomunicación, 29071 Málaga, Spain; ecasilari@uma.es

**Keywords:** fall detection system, inertial sensors, smartwatches, accelerometers, Android, battery consumption

## Abstract

Over the last few years, the use of smartwatches in automatic Fall Detection Systems (FDSs) has aroused great interest in the research of new wearable telemonitoring systems for the elderly. In contrast with other approaches to the problem of fall detection, smartwatch-based FDSs can benefit from the widespread acceptance, ergonomics, low cost, networking interfaces, and sensors that these devices provide. However, the scientific literature has shown that, due to the freedom of movement of the arms, the wrist is usually not the most appropriate position to unambiguously characterize the dynamics of the human body during falls, as many conventional activities of daily living that involve a vigorous motion of the hands may be easily misinterpreted as falls. As also stated by the literature, sensor-fusion and multi-point measurements are required to define a robust and reliable method for a wearable FDS. Thus, to avoid false alarms, it may be necessary to combine the analysis of the signals captured by the smartwatch with those collected by some other low-power sensor placed at a point closer to the body’s center of gravity (e.g., on the waist). Under this architecture of Body Area Network (BAN), these external sensing nodes must be wirelessly connected to the smartwatch to transmit their measurements. Nonetheless, the deployment of this networking solution, in which the smartwatch is in charge of processing the sensed data and generating the alarm in case of detecting a fall, may severely impact on the performance of the wearable. Unlike many other works (which often neglect the operational aspects of real fall detectors), this paper analyzes the actual feasibility of putting into effect a BAN intended for fall detection on present commercial smartwatches. In particular, the study is focused on evaluating the reduction of the battery life may cause in the watch that works as the core of the BAN. To this end, we thoroughly assess the energy drain in a prototype of an FDS consisting of a smartwatch and several external Bluetooth-enabled sensing units. In order to identify those scenarios in which the use of the smartwatch could be viable from a practical point of view, the testbed is studied with diverse commercial devices and under different configurations of those elements that may significantly hamper the battery lifetime.

## 1. Introduction

Falls suffered by the elderly are one of the most challenging problems faced by public health systems. According to the reports of the World Health Organization (WHO) [1], after road traffic accidents, falls represent the second leading cause of unintentional injury death. Adults older than 65 years are by far the population sector most sensitive to this issue, as falls may have a sizeable impact on their well-being and self-sufficiency. Data from the WHO reveal that almost a third of people aged over 65 fall each year and that this percentage climbs to 32% for those over 70 [2].

After suffering a fall, a remarkable ratio of uninjured older persons (47%) are unable to get up off the ground [3]. Moreover, lying unassisted on the floor longer than one hour after a fall is linked with a 50% mortality probability even in the absence of serious injury, as a result of co-morbidities such as pneumonia, dehydration, hypothermia, or sores [2]. In this context, the research on automatic and cost-effective Fall Detection Systems (FDSs) based on wearable devices has gained much attention in the scope of telemedicine during last decade.

FDSs are intended to differentiate common actions or ADLs (Activities of Daily Living) from movements that are suspected of being caused by falls. As soon as an accident is detected and if a local alarm is not deactivated by the monitored patient, the FDS is programmed to send an alerting message or phone call to a remote assistance provider (e.g., a relative, call center, clinical premises, and caregivers).

Wearable FDSs are built on the signals captured by sensors (mainly inertial measurement units (IMUs)) that are directly transported by the patient. In order to transmit the possible alerting messages, FDSs must be endowed with long range wireless communication interfaces (such as cell phone connections) that enable them to operate in an almost ubiquitous way, provided that the transmission system coverage is guaranteed. 

Nowadays there exist different commercial wearables specifically designed to detect falls (see, for example, the reviews presented ref. in [4,5] or [6] for an analysis of the most popular products). These off-the-shelf devices, which are typically sold in the form of a pendant or a wristband, normally incorporate a help button to call for help (a useless function if the patient remains unconscious after an accident). These alerting systems are mostly conceived for in-home monitoring through dedicated base stations with a landline connection. Apart from the cost of the detector and (in some cases) the need of long-term contracts, an additional monthly fee is required to provide cell phone service when the user demands an on-the-go (ubiquitous) tracking. Moreover, in almost all cases, the vendors do not inform about the employed detection algorithm or about the way in which the detector has been tested. Thus, the actual efficiency of these solutions to identify falls (especially when they are applied to the target population—the elderly) has not been benchmarked.

Smartphones (SP) have been considered an appealing alternative to avoid the expenses related to these specific alerting devices. However, in order to obtain an adequate characterization of the user’s mobility, the inertial sensor must be firmly attached to some point of the body (e.g., chest, waist or a limb), which implies transporting the SP in a quite unnatural position. In fact, smartphone-based FDSs may become unserviceable if the phone is carried in a bag or even in a loose shirt or trouser pocket. To cope with this problem, related literature has proposed the use of smartwatches, which may benefit from the low cost and massive popularity of this technology, originally conceived for fitness tracking. As smartphones, most smartwatches also embed in a single personal device (without requiring any bulky element that hinders the user comfort) all the hardware requirements of a wearable FDS: Inertial sensors and wireless communications. Besides, current smartwatches can be programmed and easily interfaced to other devices with a higher computation power (mainly smartphones). The placement of the watch on the wrist also enables the measurement of important biosignals (such as the pulse rate) which are also strongly affected by any accident and could be considered as an extra input signal for the fall detection algorithm. It is therefore not surprising that a “hard fall detection” function is being progressively incorporated as a native feature to some popular smartwatches such as Apple Watch (since Series 4) [7] or Samsung Galaxy Watch3 [8].

The main problem of a smartwatch based FDS is that the analysis of movements of the wrist may result in overestimates [9,10], i.e., the overabundance of false alarms caused by the jerky activity of the arms and hands, which is not always representative of the mobility of the rest of the body. The compensatory movements of the hands provokes that the wrist exhibits a completely different mobility pattern during the fall when compared to the measurements captured by other body positions [11]. As a result, when the inertial sensor is placed on the wrist, fall-related accelerometry signals may be misinterpreted with a higher probability as those originated by other ADLs and vice versa [12].

As a matter of fact, it has been shown that the best position to locate an inertial sensor aimed at characterizing the human mobility during a fall is the hip or the waist, as they are closer to the center of mass of the body [13]. Therefore, most wearables for fall detection are designed to be attached to the waist, thigh, or chest [14,15].

The classification based on a sensor on the wrist may even underperform those founded on the measurements collected on an ankle or on a knee, as it is shown in the study by Gjoreski et al. in refs. [15,16]. In these studies, it is significant that the results achieved when the watch is placed on the left arm are better than those obtained on the dominant right one. In the watch-based fall detector presented in ref. [17], authors observe that the discrimination ratio clearly improves when the watch is attached to the waist or body trunk (an illogical location for a wrist-watch).

A proper fall detection system should then incorporate a sensor located in a position less prone than the wrist to random and autonomous movements. Nevertheless, the use of the wrist in an FDS is not pointless. The combined analysis of the signals captured on the wrist and on another positions of the body may increase the efficacy of a FDS only based on the signals collected by a single sensor [18,19]. Furthermore, the signals measured by inertial sensors located on the wrist have been incorporated in several public annotated datasets (see [20] for a comprehensive review on this topic) used as benchmarking tools for fall detection architectures based on multisensory system or, at least, on multiple inertial sensors. In this regard, there is a broad consensus that sensor fusion (or the combination of the information provided by different sensors) substantially improves the efficacy of the algorithms employed for FDSs [21]. Accordingly, due to the inherent limitations of an FDS grounded on the exclusive use of a smartwatch, an approach that combines a smartwatch and one or several extra inertial units (located on other positions of the user body) should be preferred. To avoid the continuous presence of a smartphone or any other portable (Holter monitor-like) node in charge of receiving and processing the measurements from the sensors, smartwatches could assume the role of the central coordinator of this architecture with several sensing points.

Yet most smartwatches present severe restrictions in terms of battery and computing resources. In fact, autonomy together with small screens have been traditionally considered as two most relevant barriers for the adoption of smartwatches in health monitoring applications aimed at older people [22]. There is a straightforward correlation between battery drain in a smartwatch and the number of employed sensors and sample rates [23]. Thus, the battery capacity (usually more notably reduced than that of smartphones) is the most limiting factor for the deployment and adoption of applications that requires a continuous signal tracking [24].

Battery autonomy lower than 24 h can jeopardize most activity recognition systems since movement tracking would have to be suspended before bedtime for recharging the batteries. The execution of an additional fall detection (constantly running) application could seriously impact on this battery lifespan. In fact, a recent study based on questionnaires conducted with participants of a real life trial of a wearable fall detector [25], showed the users’ preferences for systems with an operational life of at least six months without recharging the battery (a duration that obviously remains far away from the actual possibilities of current smartwatches).

In ref. [26] we have shown that current commercial smartphones can be employed to deploy FDS architectures with multiple external sensors with an operational lifetime longer than 24 h provided that the use of high consumption elements (such as the screen and mobile and Wi-Fi connectivity) is minimized. In this scenario, it is legitimate to ask whether the capabilities of current smartwatches enable carrying out this role of the core of a set of portable sensing nodes (or “motes”) intended for fall detection or human activity recognition. To answer this question, we thoroughly analyze a prototype of a wearable FDS in which different models of smartwatches operate as the coordination points of a short-range wireless network of low-power inertial sensors. 

This paper is structured as follows: Section 2 reviews those existing studies that have proposed fall detection architectures that combines the measurements captured in a smartwatch with those obtained in another external sensing point. Section 3 describes the experimental testbed employed to evaluate the network prototype. Section 4 in turns presents and discusses the obtained results (mainly focused on the battery lifetime) when the prototype runs under different operational conditions. Finally, Section 5 recapitulates the main conclusions of this paper.

## 2. Related Literature 

Smartwatch-only based fall detection systems (i.e., FDSs that utilize a commercial smartwatch as the only transportable element of the system) have been analyzed in a considerable number of works (see, for example, the studies in refs. [27,28,29,30,31,32,33,34] or [35]). Conversely, the use of a smartwatch in combination with other external information sources (i.e., sensors located on positions different from the wrist) is less frequent in the related literature.

As already noted, most commercial smartwatches (or sporting smartbands, such as that employed by Li et al. in ref. [36]) are still conceived as “subsidiary” elements of the smartphones. Thus, there are several examples in the literature of these hybrid fall detection architectures that combine a smartwatch or a sporting smartband (normally used as a simple sensing point) and a smartphone. In some cases, the embedded sensors of the smartphone are also utilized by the detection algorithm. In other examples, the phone is used as the hub that receives and processes the inertial measurements of the watch to produce the detection decision. The communication between these two elements are typically carried out via Bluetooth (or Bluetooth Low Energy), since this short-range low consumption transmission technology is natively incorporated in most existing wearables.

An example of this type of architectures is the system presented by Maglogiannis et al. [37], consisting of a Bluetooth-enabled Pebble smartwatch and an Android smartphone. In order to minimize consumption, the sampling rate of the accelerometer in the watch is notably reduced (up to 5 Hz). To moderate the consumption of the wireless communications, consecutive inertial measurements are also grouped and sent every 2 s. Under these circumstances, authors report that the battery lifetime of the smartwatch can reach 30 h of constant tracking.

A similar solution is described by Vilarinho et al. in ref. [19] by connecting an LG G Watch R1 and an Android Samsung Galaxy S3 (which is transported by the user in the pocket). In the architecture, a fall is only supposed to have occurred when it is confirmed by a threshold-based analysis of the signals from the built-in accelerometers of both devices. The study of the battery consumption in the watch is again not performed.

A commodity-based smartwatch (a Microsoft band model) is Bluetooth paired with an Android smartphone to deploy a FDS in the work by Ngu et al. in ref. [38]. The detector is integrated into an Android IoT platform with cloud persistence storage and data analysis tools. Open source Weka Java package is leveraged to implement the machine learning algorithm (a Support Vector Machine). No attention is paid to battery consumption in the wearable.

An example of the use of smartphone as the central decision node and hub is provided by Ngu et al. in refs. [39] and [40]. These papers describe a hybrid FDS consisting in a machine learning classifier (a Naïve Bayes or SVM) implemented on an app running on a Nexus 5X smartphone. The detection in the phone is based on the signals received via Bluetooth from a Microsoft Band 2 smartwatch.

In this context, it has been shown [41,42] that the ratio of false alarms of FDS clearly diminishes when the detection algorithm is applied to the signals captured by both the smartphone and the smartwatch/wristband, so that the fall is only assumed if it is independently detected on both devices. Hsieh et al. have studied in ref. [41] the performance of this hybrid system depending on the position (purse, briefcase, pocket of the shirt or the pants, etc.) where the smartphone is located, concluding that the fall detection is maximized when the smartphone is located in a pocket of the pants.

A hybrid approach (combining a smartwatch and a smartphone) is also followed by Casilari and Oviedo in ref. [42]. Our initial results in that preliminary work showed that the FDS application can cause a severe decrease of the lifetime of the smartwatch battery. as more than 50% of the battery capacity (in a LG W110 G Watch R model) was depleted after 7 h of operation. Thus, the smartwatch becomes the “energy bottleneck” of the FDS. A similar conclusion is achieved by Deutsch et al. in ref. [43] after analyzing a hybrid solution consisting of a Pebble smartwatch that is Bluetooth-connected to an Android phone. The constant transmission of the measurements from the smartwatch to the smartphone resulted in the depletion of the smartwatch battery after 17–19 h.

The battery consumption of the phones in smartphone-only based FDS has been studied in some detail in some works [44,45,46,47,48,49]. However, to the best of our knowledge, a similar study with smartwatches has not been undertaken. In ref. [26], we investigated the battery drain in a hybrid smartphone-based system in which external Bluetooth low-power sensing motes are employed to transmit supplementary inertial measurements to the phone, in order to help with the movement classification performed by the detection algorithm. Now, we extend that work by substituting the smartphone by a smartwatch.

## 3. Description of the Experimental Testbed

The goal of our research is to assess the potentials of current commercial smartwatches to perform as the central node of a Body Area Network (BAN) of inertial sensors, intended to monitor human mobility. For that purpose, we deployed an experimental testbed consisting of a smartwatch and up to six IMU-enabled wireless wearable motes, which were always placed in the proximity (less than 2 m) of the watch. The nodes form a Bluetooth piconet in which the smartwatch acts as the master unit, which centralizes all the communications. As the central node of the network, the watch is responsible for communicating via Wi-Fi or mobile data to a remote monitoring point in Internet. Depending on where the role of the detector resides, we consider two generic (extreme) architectures. In the first one, the watch would process and analyze all the signals captured by the inertial sensors by implementing the corresponding fall detection algorithm. In that case, the transmissions would be limited to the alarm messages or some long-term periodical status notifications. Under the second scheme, to avoid the computing limitations of the smartwatches, the decision algorithm is assumed to be “outsourced” in an external Internet node. In that scenario, the goal of the watch would be just re-forwarding to that remote point all the signals received from the system (in an intermediate approach, the signals could be retransmitted just in case that a fall is presumed from a local preliminary analysis in the watch).

In this section we describe the characteristics of the smartwatches, the developed mobile application and the sensors employed to perform the measurements.

### 3.1. Employed Smartwatches

For the feasibility analysis, which is especially focused on the battery durability, we selected three different commercial smartwatches as a representative sample of the wide range of mid-range and high-end programmable devices currently available on the market.

Unfortunately, many smartwatches use proprietary operating systems that do not allow implementing custom applications or just tolerate slight modifications of the original native apps. Consequently, in our testbed, we only utilized Android WearOS devices as long as there exist free powerful development tools provided by Google to guarantee the control of the device’s functionalities and fully customize the application that is required for the systematic evaluation of the sensor network.

The main features of the smartwatches utilized in the testbed are displayed in Table 1.

All the employed smartwatches mount a Qualcomm Snapdragon Wear 2100 microprocessor with four 1.2 GHz Cortex-A7 cores. All of them also share the same WearOS 2.17 operating system. The most significant difference between the three models is found in the battery capacity. As can be observed from Table 1, the Skagen Falster 2 has the lower battery capacity with only 300 mAh compared to the 420 and 415 mAh of the Huawei Watch 2 and Mobvoi Ticwatch Pro 2020, respectively. On the other hand, the Mobvoi model features a bigger screen size, which could deplete the battery faster than the other smartwatches if it is not adequately configured. Finally, the three smartwatches embed an accelerometer, a gyroscope and a heart rate monitor as internal sensors. The models by Huawei and Mobvoi also integrate a magnetometer.

### 3.2. Smartwatch Application

A specific smartwatch application (running on the watch) was developed to set up the desired characteristics of the sensor network in the testbed and to record the results of the experiments. The customized app was tailored to track the battery drain of the smartwatch while operating as the core of the body sensor network. During the test, the app receives and processes all the measurements collected by the IMUs in both the external motes and, optionally, the smartwatch itself. The application was also in charge of the whole process of parameterizing the embedded sensors of the smartwatch (e.g., activation and sampling rate) and of initiating the Bluetooth piconet (i.e., detecting the presence of the motes in the vicinity of the watches and starting the corresponding connections). In particular, the application was designed to enable the user to define the following configurable elements in the testbed:The number of external motes (up to 6) in the piconet that can establish a Bluetooth connection to the watch. In this regard, the app also allows employing the internal smartwatch sensors to capture not only the inertial data but also the user heart rate.The use of the GPS information. The application is also programmable to collect (or not) the geolocation coordinates by using the GPS sensor of an associated smartphone (as the smartwatches are not capable to take those measurements even though they are supposed to implement the GPS feature). This option is intended to evaluate the impact of the GPS localization on the battery drain when the GPS service is programmed to be operative during the experiments.The exploitation of the smartwatch as a fall detector or as a simple signal gateway. As aforementioned, in a real scenario of an FDS, the smartwatch could be configured to receive and interpret the data collected by the sensors and utilize that information to feed a certain detection algorithm and decide if a fall has occurred. However, due to the complex nature of some detection techniques, the implementation of the algorithm can become unpractical for the limited resources of a smartwatch (battery, memory and computing power). A potential solution to this problem is to transfer the detection decision to an external unit (e.g., a remote Internet server). In that case, the smartwatch has to retransmit the measurements received from the sensors (the three components captured by the triaxial gyroscope, accelerometer and magnetometer and, if so configured, the heart rate) to the external server using a Wi-Fi or a 4G connection, which will also impact on the battery life.The sampling rate of the sensors. The time between two consecutive measurements of the inertial sensors in the motes can be adjusted in the app in a range from a minimum value of 10 ms (100 Hz frequency) to a maximum of 2550 ms (0.39 Hz). Unfortunately, the exact configuration of the sampling rate in the smartwatch internal sensors is not completely configurable in a reliable way as the operating system may reduce it arbitrarily if a high processing load is detected [50].

The user interface of the implemented application has been devised to take full advantage of its serviceability while minimizing its influence on the consumption. For instance, although data are continuously received from the sensors, they are only shown on the screen every thirty seconds only for feedback purposes.

When an experiment is launched, the application creates a new log or trace file including the next information:The smartwatch model.The number of sensing nodes.The selected sampling period.Two binary indicators informing whether the GPS positioning system is enabled and whether the signal retransmission is activated so that an external server is employed.A timestamp indicating the exact time in which the experiment was started.

In addition, the application also stores every two minutes in the log file information for further processing. This information consists of a timestamp, the current remaining battery level and the cumulative number of messages received and lost by the sensors employed in the experiment.

In order to avoid loss of tracking information due to the complete depletion of the battery, the application stores the last data and close the log file when it detects that the smartwatch is going to be turned off. MATLAB scripts [51] have been used to process the information stored by the log files.

### 3.3. External Sensors

The Body Sensor Network deployed for the testbed is composed by up to six Texas Instruments CC2650 SensorTag motes [52]. This small-sized Bluetooth-enabled battery-operated board suits perfectly the requirements of our study as it integrates an InvenSense MPU9250 Inertial Measurement Unit [53] that integrates a gyroscope, an accelerometer and a magnetometer.

The range of the accelerometer is selectable from four values (2 g, 4 g, 8 g, and 16 g). In the tests, a range of 4 g was set as it is normally high enough to discriminate falls from conventional ADLs [54]. In any case, the resource consumption in the smartwatch is not affected by this selection. The other sensors integrated by the SensorTags (a humidity sensor, a barometric sensor and an optical sensor) were deactivated and not employed during the experiments in order to avoid any effect on the system performance.

The SensorTag firmware defines a minimum default sampling period of 100 ms (10 Hz) for all the IMU sensors. However, this frequency is not enough to characterize the human mobility in a wearable fall detection system. In fact, a minimum sampling rate of 22 Hz has been recommended [55]). Consequently, this hard-coded limit in the SensorTag firmware was modified to enable up to a 10 ms (100 Hz) sampling period, which is the minimum resolution guaranteed by the IMU vendor. The SensorTag microprocessor was capable of managing this new sensing period because the previous commented sensors that were not used were disabled.

Every SensorTag is programmed to regularly transmit via Bluetooth to the smartwatch application the messages containing the data sensed from the inertial sensors. The original vendor firmware does not provide any mechanism to track lost messages. Hence, the SensorTag firmware had to be modified to add a sequence number to every message sent by the motes to the app, so that the lost packets could be identified by means of a simple count method.

### 3.4. Performance Metrics

A set of systematic experiments were scheduled to assess the battery durability of the smartwatch under different configuration of the testbed. For every performed test, the smartwatch was fully charged and the application was configured with a predetermined set of operational parameters, which establish the number of sensing nodes, the sampling rate, the optional use of the GPS or the inertial sensors in the wearable and the role (fall detector or gateway) of the smartwatch. During each experiment, the motes are connected to the smartwatch using a Bluetooth connection and they continuously transmit the inertial information until the smartwatch battery is drained of power. Besides, although the SensorTags can be battery operated using CR2032 coin cells, in order to prevent any malfunction caused by the battery in the motes, they were USB-powered using high capacity external batteries.

Two main factors were the initial focus of the tests: The sampling rate and the number of employed sensing motes. Therefore, we firstly studied the influence of the sampling rate on the battery consumption. Secondly, after setting a sampling frequency of 50 Hz (20 ms between two consecutive measurements), we investigated the effect of the quantity of external sensing units integrated in the BAN. Additionally, we studied the effect of enabling the internal sensors available (as an extra source of inertial data) as well as the global positioning system available in the wearable. Finally, we measured the energy cost of using the smartwatch as a simple Wi-Fi gateway between the external low-power modules and an Internet server.

For the analysis, we considered the following performance indicators:Battery life, computed as the time (in hours) from the activation of the body area net-work until wearable disconnects due to lack of power supply. Before every test, the battery was initially charged to its maximum capacity. It has to be noticed that not all the models are switched off when the battery is fully depleted but when a certain battery level is reached.Relative lifetime of the power source, estimated (in seconds/mAh) as the energy required per every unit of the battery lifetime. This metric aims at evaluating the energy cost of the network deployment regardless of the (variable) capacity of the battery of each smartwatch.Ratio (messages/mAh) between the quantity of data packets sent by the external motes (and collected at the smartwatch) and the amount of charge depleted from the battery. This indicator characterizes the power required to correctly transmit and receive every inertial measurement.Ratio (messages/mAh) between the number of lost data packets and consumed energy. This parameter identifies those networking conditions in which the BAN experience data losses.

## 4. Results and Discussion

In this section we analyze the battery duration of the smartwatches as a function of diverse operational conditions, depending on the sampling rate, the employed number of sensor nodes, the optional use of the smartwatch inertial and GPS sensors, and the alternative transmission of the captured signals to an external server.

### 4.1. Analysis of the Sampling Rate

In this section we study how the battery drain is influenced by the sampling rate. For this purpose, we measured the battery lifetime (until the smartwatch is switched off) when the data sampling period is modified from 10 ms (100 Hz), which is the minimum value supported by SensorTag node, to 80 ms (12.5 Hz). The experiments were initially performed using a single external SensorTag. Results for the four considered performance metric and for the three smartwatches under test are represented in Figure 1.

Figure 1a shows that the battery duration highly depends on the sensing period but also on the smartwatch model. The Skagen Falster 2 model exhibits the lowest durability as expected because this smartwatch has the lowest battery capacity (300 mAh). However, the Huawei Watch 2 and the Mobvoi TicWatch Pro models, which have similar capacity, present a very different battery lifetime, with the Huawei Watch 2 achieving a lifespan between 35% and 45% longer than that of the Mobvoi TicWatch Pro. This discrepancy can be attributable to two factors. Firstly, the Mobvoi TicWatch Pro screen has a larger size (1.39 inches in contrast to 1.2 inches of the Huawei Watch 2), which straightforwardly increases the battery consumption, and, secondly, the Huawei Watch 2 is capable of entering in a low-power screen mode without ending the wireless communications. In this regard, the implemented app running in the smartwatches benefits from the WearOS ambient mode, which is a low-power state that maintains the application constantly visible to the user but setting the screen illumination to the minimum while keeping active the rest of the functionalities in the smartwatch that are required to support the connections to the motes. This mode had to be activated because as soon as a smartwatch detects that the user is not interacting with the application during a certain time, the screen is turned off and the wireless connections are automatically closed in order to reduce the battery consumption. As it was expected, Figure 1a also reveals that the battery duration clearly decreases when the sensing period is reduced, since the amount of data to be processed and transmitted augments. In particular, the use of the highest data rate (100 Hz) provokes a reduction of the battery duration of up to 35% with respect to the cases under the lowest sampling rate for the Huawei and Skagen model. Similarly, for the Mobvoi TicWatch this diminishment of the durability climbs to 45%. A similar conclusion is obtained from the study of the curve of the relative lifetime, which is depicted in Figure 1b. This metric does not take into consideration the absolute battery capacity but the battery duration per energy unit (mAh). From this figure we conclude that, in relative terms, the Skagen Falster 2 smartwatch is less sensitive to the sensing period when compared with the other smartwatches as the increase of the battery consumption is even softened for the highest frequency rates.

Figure 1c in turn illustrates the evolution of the useful information (expressed in terms of the number of data messages arriving at the watches) received per unit of depleted energy when the sensing period is modified. Predictably, the curves follow a reciprocal behavior to that exhibited by the lifetime as a lower sampling period implies a higher amount of received messages. The Figure shows that the election of the smartwatch model noticeably may impact on the energy cost of conveying a data packet. Thus, the Huawei Watch 2 model is capable of receiving the highest number of messages per mAh for all the considered sensing periods. On the other hand, the Skagen Falster 2 and the Mobvoi TicWatch Pro present a similar behavior, especially for low sensing periods.

Finally, Figure 1d shows the total number of lost messages per consumed mAh as a function of the sensing period. As observed in the figure, losses are zero for all the sensing periods except for the lowest sampling period of 10 ms (which corresponds to the highest sampling rate of 100 Hz). For that frequency, the Huawei Watch 2 and the Mobvoi TicWatch Pro respectively present almost 25 and 7 lost messages per energy unit. However, those results can be considered acceptable (even for the Huawei Watch 2) as the number of lost messages accounts only for 0.25% of the total number of transmitted messages.

For the rest of the experiments in the following sections, we set the sampling rate to 50 Hz (sensing period of 20 ms), a value which simultaneously minimizes the transmission losses and fulfills the requisites for an adequate sampling rate for fall detection purpose. In this vein, there are studies in the literature (see, for example [55,56]) that have shown that a sampling rate between 20 and 40 Hz is enough to properly characterize the human mobility in a wearable system aimed at detecting falls. In fact, 50 Hz is the most common sampling frequency utilized by the related literature about FDSs [57].

Regarding the short-term energy consumption patterns in the models under test, notable divergences can be found. By way of example, Figure 2 represents the current drained from the smartwatch battery during three minutes of execution of the application after the initial setup of the connection with the sensing mote. The figure apparently shows very different behaviours depending on the smartwatch model. The Skagen Falster 2 and Mobvoi TicWatch Pro models have more stable battery consumption with occasional peaks, while the Huawei Watch 2 displays a more erratic behaviour. In any case, Table 2 shows that the mean long-term current consumption is almost equal for the Huawei Watch 2 and the Mobvoi TicWatch Pro and slightly lower for the Skagen Falster.

From a practical point of view, an FDS could be considered “autonomous” if it can seamlessly work for at least a whole day without having to be removed or disconnected for a battery recharge. Thus, a minimum battery autonomy of sixteen hours (a whole day excluding eight sleep hours) is essential for the viability of any fall detection application. In this sense, only the Huawei Watch 2 is able to work for such amount of time but only at the cost of using a short sampling period (80 ms), which cannot be sufficient for an adequate identification of the brusque movements provoked by falls. Thus, a reduced battery capacity, such as that of the Skagen Falster 2, may become a not negligible impediment for the deployment of realistic smartwatch-based FDS with an external sensing unit.

To evince that this conclusion can be extended to other existing commercial smartwatches (not analyzed in our testbed), Table 3 recapitulates the battery capacity of a long list of WearOS smartwatches. The data in the table (obtained from the corresponding vendors) is sorted from the lowest to the highest battery size. The smartwatches employed in our study are highlighted in bold. This table shows that, at the time of writing this paper, there are not many smartwatches with a battery capacity significantly higher than those integrated in Huawei Watch 2 and Mobvoi TicWatch Pro 2020 models, except for the case of Mobvoi TicWatch 3, which features 41% larger battery capacity.

### 4.2. Impact of the Number of Sensors

For the next experiment, we modified the number of wireless sensors connected to the smartwatches. As noted before, we fixed a sampling period of 20 ms (50 Hz) and then measured the battery consumption when up to 6 Bluetooth SensorTags are simultaneously connected to the smartwatch to transmit their inertial measurements.

Figure 3a,b respectively show again the battery duration in absolute terms and the lifetime (seconds) per depleted energy unit (mAh) as a function of the number of connected motes (ranging from only one to six external devices). From the figures, it can be observed that the battery depletion is clearly influenced by the number of active Bluetooth connections to the sensing nodes. In this way, the battery duration using six motes is half of that obtained when just one SensorTag is present in the Bluetooth piconet coordinated by the smartwatch. In a previous study, we detected a similar behavior of the battery when a smartphone acts as the central node (master) of the Bluetooth piconet [26]. In this case, Figure 3 illustrates that the three tested smartwatch models present the same decay of the lifetime, thereby reducing the appropriateness of the smartwatches to deploy an FDS in those scenarios where several inertial units are necessary.

In the performed experiments, the Skagen Falster 2 and the Mobvoi TicWatch 2020 present a very similar relative battery duration (Figure 3b) for all the tested number of sensors, although the Mobvoi TicWatch 2020 is able to be operative during a longer period (Figure 3a) as it is provided with a battery 38% larger than that of the Skagen Falster 2. The battery duration of the Huawei Watch 2 outperforms that of the other models for all the configurations of the piconet.

Figure 3c depicts the energy cost of transmitting messages from the motes to the smartwatch. The figure follows the same trends commented for Figure 3b as the Huawei Watch 2 is able to receive the highest number of messages per consumed energy unit for the six considered topologies, followed by the Mobvoi TicWatch Pro and the Skagen Falster 2. Finally, Figure 3d represents the number of lost messages per mAh as a function of the number connected motes.

There is not a pattern for any smartwatch that relates the number of lost messages with the number of connected sensors. The figure shows that the use of multiple connections (due to the existence of several Bluetooth slaves) can cause occasional instabilities in the connections that cause some packet losses (in this regard, no specific or common loss pattern was detected). Nevertheless, the number of lost messages (even for the configuration with six external nodes) is very low (less than 0.1%) when compared to the number of correctly received data packets. In any case, these results confirm the capability of the commercial smartwatches to simultaneously maintain multiple simultaneous BLE connections to several external devices in an efficient manner.

### 4.3. Influence of the Use of Internal Sensors

In the previous experiments the functionality of the smartwatch was limited to the networking plane, i.e., it performed just as the central node (core) of the wireless BAN without activating its internal embedded sensors. In an actual application scenario of a smartwatch-based FDS, the smartwatch (as an additional sensing point located on the wrist) should also capture those signals that help to produce the decision of the detection algorithm.

In order to evaluate the impact on the consumption of using the sensors embedded in the smartwatch, we repeated the precedent tests but now activating the internal IMU of the watch. Thus, the triaxial measurements of the accelerometer, gyroscope, and magnetometers captured on the wrist were also logged in parallel with those received via Bluetooth from the external units.

Additionally, aiming at assessing the influence of an embedded biosignal sensor, we replicated the experiments when the Heart Rate (HR) monitor of the smartwatches is alternatively turned off or turned on. In the tests the HR monitor is configured to report a measurement every second (1 Hz). As for the internal IMU, a sampling frequency of approximately 50 Hz was achieved by setting the rate to the SENSOR_DELAY_GAME constant value in the Android sensor manager of the watches [58].

The results for the three smartwatch models are respectively graphed in Figure 4a–c. The figures disclose the dramatic effect of the activity of the inertial sensor on the life of the smartwatches’ battery, especially when a low number of external units are employed. In particular, when just a single SensorTag is present, the battery duration is halved when the internal IMU is activated. Thus, under the simplest configuration of the detection piconet (the smartwatch plus and an external node) the battery lifespan is reduced to less than 3.5 or 5 h (depending on the model). In the case of Huawei Watch 2 model, this tendency is aggravated (with a supplementary reduction of the battery lifetime of about 20%) when the HR monitor is activated while a minor repercussion is reported for the other two models when the heart rate is measured.

In the results obtained in the following sections, the internal sensors of the watches are switched off.

### 4.4. Impact of GPS Usage

In order to expedite assistance in an emergency situation, an FDS should provide a positioning system, which can be especially useful when the user loses consciousness after falling or when monitoring takes place outdoors. By supporting this feature, the FDS is enabled not only to detect the fall but also to inform the family or the medical services about the location where the fallen user can be found and assisted. Most mid- and high-end watches natively integrate GPS location, so this service can be easily incorporated into a smartwatch-based FDS. However, the extensive use of a “power hungry” service, such as the built-in GPS sensor, may dramatically affect the energy drained by the wearables [59]. In order to minimize this consumption, in a practical implementation of an FDS, GPS service could be switched on only when an alerting message has to be transmitted. However, this configuration could increase the time of the alarming system due to the startup time needed by GPS chip to adjust its clock with the signals received from the GPS satellites.

To assess the effects of the geolocation capability on our prototype, we have measured the battery consumption in the smartwatches when the GPS sensor is both activated and deactivated. To emulate a realistic energy-demanding scenario in which the location information is frequently updated, we have forced the watches to request the GPS position on a periodic basis with a moderate value of the sampling period. In particular, the watches read the GPS sensor every ten seconds (an interval short enough for a proper location of a walking subject). During all the experiments the whole system (sensors and watches) remained motionless.

Figure 5 depicts the results (in terms of battery duration) of using or not the GPS positioning system as a function of the number of employed sensing nodes. Although GPS consumption could be influenced by other factors, such as the strength of the signal received from the satellites [60], graphs show that the use of GPS has a marginal impact on the battery lifetime, especially when it is compared with the effect of increasing the number of connected sensors. This conclusion can be applied to the three smartwatch models under study, although a slightly higher reduction of the battery duration in detected in the Mobvoi TicWatch Pro.

### 4.5. Influence of Using an External Server

The computing and memory resources required by a wearable FDS heavily depend on the complexity of the detection algorithm used to discriminate falls from ADLs. Some of those algorithms, especially those comprising artificial intelligence methods, may not be suitable to be executed locally in a smartwatch due to the limited computational ability of this type of wearables. In those cases, it may be necessary to “outsource” the core decision system of the FDS by implementing the fall detection algorithm in an external (e.g., Internet-connected) server. This remote server would be in charge of producing the detection decision based on the inertial data captured by the Bluetooth-enabled sensing motes. Thus, under this architecture, the watch should operate as a gateway-like node, responsible for retransmitting to the server (via Wi-Fi or a 3G/4G cellular connection) the information received from the sensors in the Bluetooth piconet. As the permanent use of a wireless connection may deplete the battery rapidly, we evaluated the influence of the use of the smartwatch as a re-transmitter of the sensed data. For that purpose, we repeated the previous experiments but now forcing the wearable to re-send all the received inertial sensed data to an external Internet server by means of a Wi-Fi connection (cellular interfaces are less common in smartwatches and may cause a higher consumption [61]). To perform the tests, a specific data server was programmed and deployed in a laptop, which was connected to the same wireless local area network of the smartwatches. At the beginning of each trial, the smartwatch opens a TCP/IP socket connection to the server which is employed to send the inertial data during the whole experiment (until it is closed when the full battery depletion is detected).

The system was evaluated under two configurations: For the first one, the watches retransmitted continuously, that is to say, the data packets collected from the sensors were sent as soon as they were received. In the second one, the smartwatches stored, retained and grouped the received messaged to retransmit them periodically. Different sending periods were considered. Figure 6 shows the battery duration and relative battery lifespan for the case of using a sampling rate of 50 Hz with an external sensing mote. Graphs illustrate the metrics as a function of the sending interval at which the periodical Wi-Fi retransmission take place (15, 20, 30, and 60 s, which correspond to a sending frequency ranging from 1 to 4 retransmissions per minute). In the figures, the continuous retransmission of the data is represented in those points corresponding to a null interval. In the last point of each curve, for comparison purposes, we have also represented the battery duration when the Wi-Fi connection is not enabled (and the detection decision is supposed to be locally generated by the watches).

As expected, the graphs reveal the high impact of using a Wi-Fi connection. In particular, the battery duration is halved for the Mobvoi TicWatch Pro and reduced by a factor of three in the other models. Graphs also show that the configuration of the Wi-Fi retransmission and the sending period do not affect the battery depletion significantly, although a continuous data stream transmission may provoke a slight decrease of the battery duration (except for the case of Huawei Watch 2). In view of these results, the use of Wi-Fi in a smartwatch-based FDS is not recommendable.

## 5. Conclusions

Smartwatches have been proposed by different studies as a cost-effective solution to deploy wearable fall detection systems. Smartwatch-based FDSs can benefit from the decreasing costs and widespread popularity of these personal monitoring devices as well as from the inertial and biosignal sensors that are natively integrated in these wearables. However, due to the particular dynamics of the wrist (whose movements are not always correlated with the activity of the rest of the body), a FDS merely based on the measurements captured by a smartwatch may be prone to errors (e.g., false alarms caused by jerky or a sudden activity of the arms). To cope with this problem, extra specific low-cost low-range wireless sensors could be placed in other body positions to complement the measurements collected by the watch. To avoid the presence of an extra “sink” or central node (such a smartphone, which is not always in the vicinity of the users), these external nodes could directly communicate with the smartwatch, which could implement a fall detection algorithm fed by the data measured by the different nodes of this Body Area Network.

In this paper, we have empirically investigated the feasibility of this wearable architecture devised for fall detection. For this purpose, we set up a testbed consisting of a Bluetooth piconet based on a smartwatch (up to three different commercial models were alternatively considered) and up to six external low-power Bluetooth-enabled sensing motes provided with an Inertial Measurement Unit (embedding an accelerometer, a gyroscope, and a magnetometer).

The operational impact of deploying this networking solution on a smartwatch was thoroughly analysed under different configuration of the prototype. In particular, we assessed the battery lifetime depending on a variety of factors: The number of external nodes composing the star topology of the BAN, the sampling frequency at which the measurements are collected and transmitted to the watch, the activation (or not) of the internal IMU and heart rate sensor of the watches, the use of GPS and the role of the smartwatch in the architecture (as the final decision node or as an intermediate hub that communicates with an external decision point in charge of analysing the measurements).

Except for the case of the usage of the GPS, the performed tests show that all these factors have a remarkable effect on the battery drain. For the most favorable configuration (just one external sensing node, no use of the internal sensors, no Wi-Fi connections to any external server and a very low sampling rate-lower than 15 Hz-), none of the analyzed models exhibit a battery lifetime longer than 20 h. Tests show that this duration rapidly diminishes if we increase the number of external nodes or the sampling frequency. For example, with 50 Hz, which is far more adequate than 15 Hz to characterize the human mobility during falls, battery duration is shortened by more than 30%. Similarly, the battery life is shortened to 2–4 h when the internal inertial unit of the watches is employed or when the smartwatch just assumes the role of a data gateway and the measurements received from the external motes are constantly retransmitted via Wi-Fi to an external powered node where the fall detection could be implemented without computing or memory restrictions. This short lifespan could be completely unacceptable for any practical application of a wearable BAN, given that the application should be interrupted to recharge the battery several times a day.

Future studies should confirm these results with other popular smartwatch models and operating systems, in particular Apple Watch and iOS devices, which are still by far the high-end model with the highest market share [62]. However, if we take into account the typical battery capacity of most commercial smartwatches (similar to those employed in our testbed), we can conclude that the battery life of current smartwatches still remains a noteworthy limiting factor to deploy networking architectures intended for fall detection.

In addition, we also should consider that the implementation of most effective algorithms to detect falls (normally based on artificial intelligence methods and complex deep learning techniques), which have been proposed by the related literature, still pose a significant challenge to the scarce computing and memory resources of these popular wearables.

In any case, the social acceptability of smartwatches among the elderly is a key element to foster its use as a monitoring tool. Some studies [63] have revealed that older people are still reluctant to see smartwatches as a credible alternative to sensitive issues related to aging (such as managing emergencies or tracking health indicators).

## Figures and Tables

**Figure 1 sensors-21-02254-f001:**
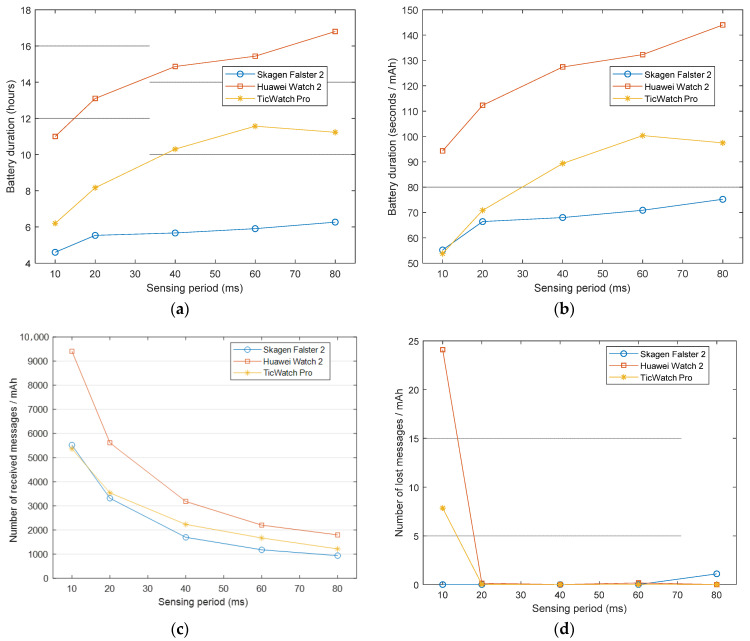
Performance metrics of the smartwatches under test vs. data sampling period (ms): (**a**) Absolute battery lifetime (in hours), (**b**) relative battery duration (lifespan per depleted energy unit), (**c**) number of messages received per depleted energy unit, (**d**) number of lost messages per depleted energy unit.

**Figure 2 sensors-21-02254-f002:**
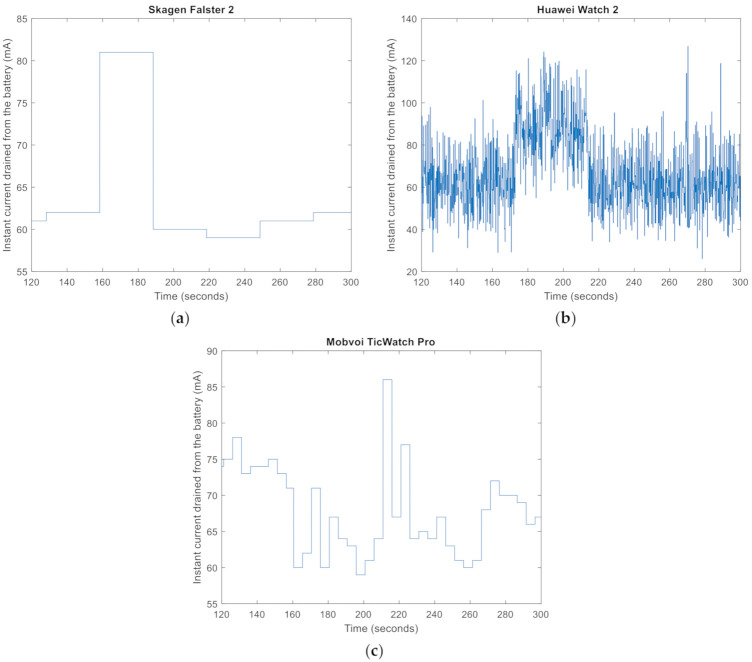
Snapshot of the evolution of the current drained for the Skagen Falster 2 (**a**), Huawei Watch 2 (**b**) and Mobvoi TicWatch Pro (**c**) during three particular minutes (from second 120 to 300) of the experiment with a frequency sample of 50 Hz and one connected sensing node.

**Figure 3 sensors-21-02254-f003:**
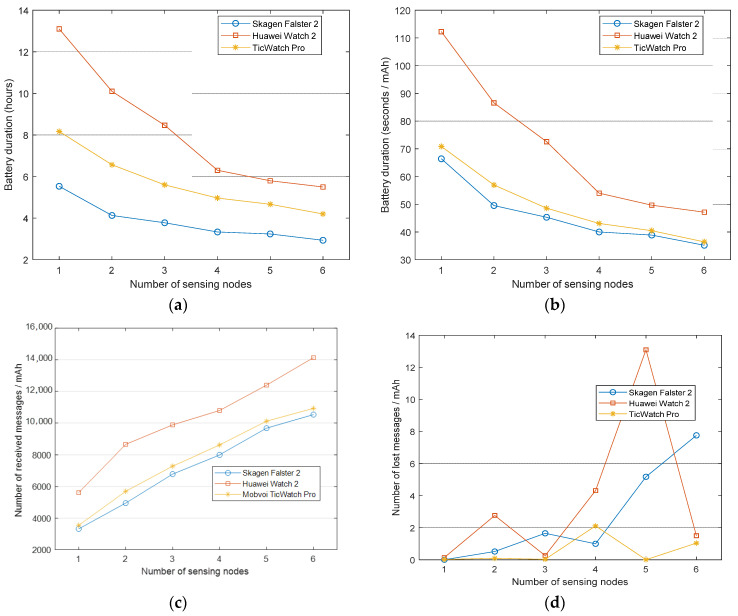
Performance metrics as a function of the number of connected sensors for a sampling rate of 50 Hz: (**a**) Battery duration, (**b**) relative battery lifetime (lifespan per consumed mAh), (**c**) number of messages received per consumed mAh, and (**d**) number of lost messages per consumed mAh.

**Figure 4 sensors-21-02254-f004:**
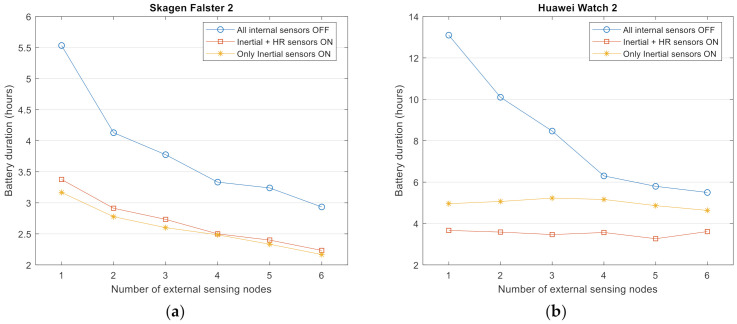
Measured battery duration as a function of the number of (external) connected sensing nodes for a sampling rate of 50 Hz when the internal sensors of the smartwatch (inertial measurement unit (IMU) and heart rate monitor) are all connected (ON), all disconnected (OFF) and when only the inertial sensors (IMU) are connected: (**a**) Results for Skagen Falster 2, (**b**) Results for Huawei Watch 2, (**c**) Results for Mobvoi TicWatch Pro.

**Figure 5 sensors-21-02254-f005:**
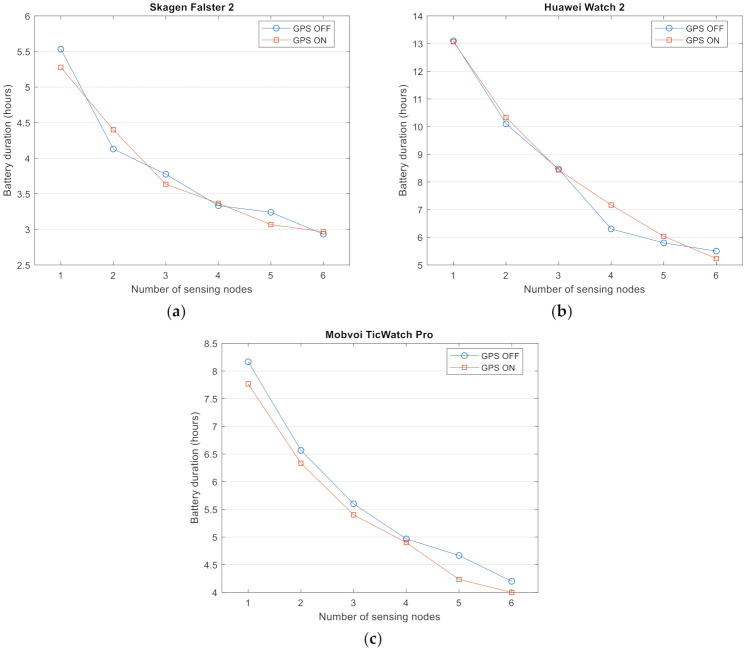
Battery duration as a function of the number of connected sensors for a sampling period of 20 ms (sampling rate of 50 Hz) depending on the use of the localization services for the different smartwatches: (**a**) Skagen Falster 2, (**b**) Huawei Watch 2, and (**c**) Mobvoi TicWatch Pro.

**Figure 6 sensors-21-02254-f006:**
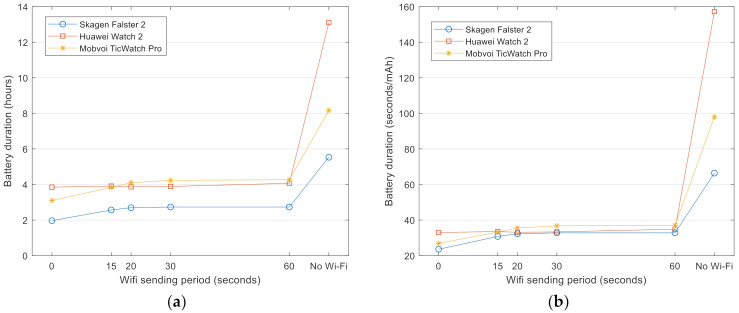
Battery life as a function of the period utilized to retransmit the inertial data to the external server via Wi-Fi (the 0 period indicates that the data is sent continuously): (**a**) Battery duration (**b**) relative battery lifetime (lifespan per consumed mAh).

**Table 1 sensors-21-02254-t001:** Characteristics of the smartwatches employed in the testbed.

Characteristics	Huawei Watch 2	Skagen Falster 2	Mobvoi Ticwatch Pro 2020
Release date	2017	2018	2019
Screen size (inches)	1.2	1.2	1.39
RAM (MB)	768	512	1024
Bluetooth version	4.1	4.1	4.2
Battery capacity (mAh)	420	300	415
Internal sensors	Accelerometer, Gyroscope, Magnetometer, Heart Rate Monitor	Accelerometer, Gyroscope, Heart Rate Monitor	Accelerometer, Gyroscope, Magnetometer, Heart Rate Monitor

**Table 2 sensors-21-02254-t002:** Mean instant current (mA) consumed using a sensor mote and a sampling frequency of 50 Hz.

	Skagen Falster 2	Huawei Watch 2	Mobvoi TicWatch Pro
Mean current (mA)	64.12	67.78	67.83

**Table 3 sensors-21-02254-t003:** Comparison of the battery capacity (in mAh) of popular WearOS smartwatches.

Smartwatch	Battery Capacity (mAh)	Smartwatch	Battery Capacity (mAh)
Polar Ingnite	165	Armani Connected	364
Armani Exchange Connected	300	Michael Kors Access Grayson	370
Diesel Full Guard 2.5	300	Mobvoi TicWatch C2	400
Hublot Big Bang E	300	Tag Heuer Connected Modular 45	410
Louis Buiton Tambour Horizon	300	Mobvoi TicWatch E2	415
Michael Kors Access Sofie	300	Mobvoi TicWatch S2	415
Misfit Vapor X	300	**Mobvoi TicWatch Pro 2020**	**415**
Mobvoi TicWatch S & E	300	Sony Smartwatch 3	420
Montblanc Summit	300	**Huawei Watch 2**	**420**
OPPO Watch	300	Tag Heuer Connected	430
Kate Spade Scallop Smartwatch 2	300	LG Watch Sport	430
**Skagen Falster 2**	**300**	Montblanc Summit 2+	440
Fossil Q Julianna HR & Fossil Gen 5	310	Huawei Watch GT 2	445
Skagen Falster 3	310	Suunto 7	450
Misfit Vapor 2	330	Huawei Watch GT 2 Pro	455
Montblanc Summit 2	340	Huawei Watch GT 2e	455
Tag Heuer Connected Modular 41	345	Polar M600	500
Fossil Sport	350	Mobvoi TicWatch 3	595
Moto 360	355		

## Data Availability

Data is contained within the article.

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
