# Peer review of "A Feasibility Study of the Use of Smartwatches in Wearable Fall Detection Systems"

_sensors, 2021, doi:10.3390/s21062254_

Round 1

Reviewer 1 Report

I think you have tackled the problem of battery lifetime in a structured and good way. You also points out the problem with new sensors and applications for the Smartwatches when the battery lifetime is a limit factor. The method and how you describe methods and results I think is fine.

The sections 1 and 2 have a lot of text and the same facts are mentioned several times with or without references. All facts shall have references. Especially section one need to be rewritten but also section 2. Focus of what is needed for fall detection and what have been done before regarding battery consumption. Do not wright a lot of other things in the manuscript. Also a research question, aim or hypothesis must be mentioned in the first chapter. Finally in chapter 3 you mention what the paper is about.

  1. Description of the Experimental Testbed. I just have some small comments:

Line 473 It is better to put future work in conclusion or discussion not in method.

Line 483 and 488 four! 3?

Line 556 Missing a dot. Have seen that in some other line also but can´t find it again.

Line 557-559 need a reference.

  1. Results and discussion I just have some small comments:

Line 645 Tichwatch use Mobvoi Ticwatch

Line 676 Don´t use several and only have 2 references.

Author Response

I attach a file with the reply.

Thanks.

Reviewer 2 Report

Well written and clear paper.

It could be of large interest to those interested in depicting scenarios aimed to increase the usability of fall detection systems.

Minor spelling errors:

Line 111: shirt o (should be shirt "or")

line 188: than should be that

line 220: an especial should be a special or a particular

line 472 : an should be and

line 566: recommend should be recommended

Author Response

I attach a file with the reply.

Thanks.

Author Response

I attach a file with the reply.

Thanks.
